# Neutralizing Activity of Sera from Sputnik V-Vaccinated People against Variants of Concern (VOC: B.1.1.7, B.1.351, P.1, B.1.617.2, B.1.617.3) and Moscow Endemic SARS-CoV-2 Variants

**DOI:** 10.3390/vaccines9070779

**Published:** 2021-07-12

**Authors:** Vladimir A. Gushchin, Inna V. Dolzhikova, Alexey M. Shchetinin, Alina S. Odintsova, Andrei E. Siniavin, Maria A. Nikiforova, Andrei A. Pochtovyi, Elena V. Shidlovskaya, Nadezhda A. Kuznetsova, Olga A. Burgasova, Liudmila V. Kolobukhina, Anna A. Iliukhina, Anna V. Kovyrshina, Andrey G. Botikov, Aleksandra V. Kuzina, Daria M. Grousova, Amir I. Tukhvatulin, Dmitry V. Shcheblyakov, Olga V. Zubkova, Oksana V. Karpova, Olga L. Voronina, Natalia N. Ryzhova, Ekaterina I. Aksenova, Marina S. Kunda, Dmitry A. Lioznov, Daria M. Danilenko, Andrey B. Komissarov, Artem P. Tkachuck, Denis Y. Logunov, Alexander L. Gintsburg

**Affiliations:** 1Federal State Budget Institution “National Research Centre for Epidemiology and Microbiology Named after Honorary Academician N F Gamaleya” of the Ministry of Health of the Russian Federation, 123098 Moscow, Russia; shchetinin.alexey@yandex.ru (A.M.S.); avocnido.anila@gmail.com (A.S.O.); andreysi93@ya.ru (A.E.S.); marianikiforova@inbox.ru (M.A.N.); a.pochtovyy@gmail.com (A.A.P.); lenitsa@gmail.com (E.V.S.); nadyakuznetsova0@gmail.com (N.A.K.); olgaburgasova@mail.ru (O.A.B.); lkolobuchina@yandex.ru (L.V.K.); sovanya97@yandex.ru (A.A.I.); annkovyrshina@gmail.com (A.V.K.); tessey@mail.ru (A.G.B.); a.kuzina.v@gmail.com (A.V.K.); dgrousova@gmail.com (D.M.G.); amir_tuhvatulin@yahoo.com (A.I.T.); sdmitryv@yahoo.com (D.V.S.); olga-zubkova@yandex.ru (O.V.Z.); olv550@gmail.com (O.L.V.); rynatalia@yandex.ru (N.N.R.); aksenova16@gmail.com (E.I.A.); markunda99@gmail.com (M.S.K.); artem.p.tkachuk@gmail.com (A.P.T.); gintsburg@gamaleya.org (A.L.G.); 2Department of Virology, Biological Faculty, Lomonosov Moscow State University, 119991 Moscow, Russia; 3Department of Molecular Neuroimmune Signalling, Shemyakin-Ovchinnikov Institute of Bioorganic Chemistry, Russian Academy of Sciences, 117997 Moscow, Russia; 4Moscow Healthcare Department, 127006 Moscow, Russia; ovk_67@mail.ru; 5Department of Infectious Diseases, Peoples’ Friendship University of Russia (RUDN University), 117198 Moscow, Russia; 6Smorodintsev Research Institute of Influenza, 197022 St. Petersburg, Russia; dlioznov@yandex.ru (D.A.L.); daria.danilenko@influenza.spb.ru (D.M.D.); a.b.komissarov@gmail.com (A.B.K.); 7Department of Infectious Diseases and Epidemiology, First Pavlov State Medical University, 197022 St. Petersburg, Russia; 8Department of Infectiology and Virology, Federal State Autonomous Educational Institution of Higher Education I M Sechenov First Moscow State Medical University of the Ministry of Health of the Russian Federation (Sechenov University), 119435 Moscow, Russia

**Keywords:** COVID-19, SARS-CoV-2, vaccine, Sputnik V, VOC, virus neutralizing activity

## Abstract

Since the beginning of the 2021 year, all the main six vaccines against COVID-19 have been used in mass vaccination companies around the world. Virus neutralization and epidemiological efficacy drop obtained for several vaccines against the B.1.1.7, B.1.351 P.1, and B.1.617 genotypes are of concern. There is a growing number of reports on mutations in receptor-binding domain (RBD) increasing the transmissibility of the virus and escaping the neutralizing effect of antibodies. The Sputnik V vaccine is currently approved for use in more than 66 countries but its activity against variants of concern (VOC) is not extensively studied yet. Virus-neutralizing activity (VNA) of sera obtained from people vaccinated with Sputnik V in relation to internationally relevant genetic lineages B.1.1.7, B.1.351, P.1, B.1.617.2, B.1.617.3 and Moscow endemic variants B.1.1.141 (T385I) and B.1.1.317 (S477N, A522S) with mutations in the RBD domain has been assessed. The data obtained indicate no significant differences in VNA against B.1.1.7, B.1.617.3 and local genetic lineages B.1.1.141 (T385I), B.1.1.317 (S477N, A522S) with RBD mutations. For the B.1.351, P.1, and B.1.617.2 statistically significant 3.1-, 2.8-, and 2.5-fold, respectively, VNA reduction was observed. Notably, this decrease is lower than that reported in publications for other vaccines. However, a direct comparative study is necessary for a conclusion. Thus, sera from “Sputnik V”-vaccinated retain neutralizing activity against VOC B.1.1.7, B.1.351, P.1, B.1.617.2, B.1.617.3 as well as local genetic lineages B.1.1.141 and B.1.1.317 circulating in Moscow.

## 1. Introduction

The recent successful launch of SARS-CoV-2 vaccines gives hope for an early reduction of the pandemic and a return to the pre-quarantine living conditions [1,2,3,4,5]. All major vaccines ensure a convincing level of protection (over 90%) in the short term and reliable protection against the severe course of COVID-19 according to the clinical trials results. For countries leading the universal immunization program, the downward trend in disease incidence and mortality is evident [6]. The statistics data for Israel, the UAE, the USA, and the UK show a sharp decline in morbidity and mortality from COVID-19 after reaching 30 doses per 100 people [7]. It is not clear how a similar level of protection for widely used vaccines can be maintained for new SARS-CoV-2 strains with mutations in the SARS-CoV-2 Spike (S; envelope glycoprotein). Since most vaccines include Spike as a principal immunogen, its variability surveillance can timely inform on the risks of escape from neutralizing antibodies formed by vaccination.

Currently, over a million of SARS-CoV-2 genomic sequences are available on the GISAID server [8]. Individual virus variants with more pronounced epidemiological, immunological, or pathogenic properties, are of concern. At the beginning of 2021, the list of variants of concern (VOC) included the B.1.1.7, B.1.351, P.1; currently, this list already contains ten records including linages B.1.617.1, B.1.617.2, B.1.617.3 [9]. For some VOC, identical mutations in S proteins appear independently, requiring in-depth research into their effects on transmissibility, severity, and ability to overcome immunity formed in convalescents and vaccinated.

Active study of the composition of the mutations that have been accumulating recently in S protein shows that most of them increase interaction with the host’s ACE2 receptor [10]. The neutralizing effect of serum samples obtained from patients vaccinated with Pfizer/BNT162b2 was reduced for the B.1.351, B.1.1.28, and B.1.617 variants by 7.85, 5.12, and 3 times, respectively, in comparison to the wild-type virus [11,12]. Assessment of individual mutation contributions shows the major impact of the E484K mutation [11,13]. A comparative study of the decrease in neutralizing activity for serum samples obtained from the patients vaccinated with Moderna/mRNA-1273 and those vaccinated with Pfizer/BNT162b2 showed 20- and 40-fold decrease respectively against B.1.351 [14]. In fact, the cross-neutralization of the B.1.351 variant was comparable to SARS-CoV-1 and Bat SL-CoV-WIV1, suggesting that a relatively small number of Spike mutations may result in the escape from neutralizing antibodies. It is becoming apparent that mutations in RBD can pose the greatest risk, both of making SARS-CoV-2 more contagious and of reducing antibody neutralization. These mutations are likely to include K417N, L452R, E484K, S494P and N501Y/T, based on molecular dynamics data [15].

We have previously developed and tested the Sputnik V vaccine, which forms the high titers of neutralizing antibodies and a profound cell immune response [5]. The protective effectiveness of the vaccine was 91.6% as assessed by the results of phase III clinical trials conducted in Russia. Sputnik V vaccine is currently available for use in more than sixty-two countries. Tens of millions of people received the vaccine in Russia and around the world. This study was aimed to investigate the virus-neutralizing activity of sera for “Sputnik V”-vaccinated against VOC (B.1.1.7, B.1.351, P.1 and B.1.617) as well as dominant genetic lineages in Moscow with substitutions in the RBD.

## 2. Materials and Methods

### 2.1. Analysis of SARS-CoV-2 Genetic Diversity in Russia

We downloaded all available metadata from GISAID (accessed on 16 April 2021) querying keyword “Russia” in the Location field. The dataset was filtered as follows: 8 sequences were removed due to the incomplete sample collection date, 3 sequences—due to the sample collection date in March 2021 in order to avoid skewing the relative frequencies of genotypes and mutations during that month, and 25 sequences—due to the missing genotype information. Amino acid substitution frequencies were analyzed from the corresponding column of the metadata with Pandas library v.1.2.1 and Matplotlib library v.3.3.3 in Python v.3.7.8 using one-hot coded mutations and calculated means on monthly resampled data. Mutations spanning 319 to 541 positions of Spike were considered as mutations in RBD.

### 2.2. Sampling and Identification of SARS-CoV-2

Sequencing of SARS-CoV-2 Moscow diversity was carried out in a frame of collaboration with Moscow City Infectious Disease Clinical Hospital No.1 Moscow Healthcare Department. The study was approved by the ethics committee (the Local Ethics Committee Protocol No. 2a of 11 May 2020, No. 11a of 16 November 2020, and No. 1 of 11 February 2021). Nasopharyngeal swab was collected from the patients involved in the study for further examination by quantitative reverse transcription PCR. SARS-CoV-2 RNA was identified in the samples using a reagent kit for extraction and qualitative determination of SARS-CoV-2 coronavirus RNA using the SARS-CoV-2 FRT RT-PCR method, manufactured by Gamaleya National Research Institute of Epidemiology and Microbiology. Virus isolation and sequencing were carried out in separate aliquots.

### 2.3. Cell Lines and Viruses

Vero E6 (ATCC CRL-1586), 293T, and 293T/ACE2[16] cells were maintained in complete Dulbecco’s modified Eagle’s medium (DMEM), containing 10% fetal bovine serum (FBS, HyClone|Cytiva, Logan, UT, USA), L-glutamine (4 mM) and penicillin/streptomycin solution (100 IU/mL; 100 μg/mL) (PanEco, Moscow, Russia).

SARS-CoV-2 strains B.1.1.1 or PMVL-1 (GISAID EPI_ISL_421275), B.1.1.141 (T385I) or PMVL-31 (GISAID EPI_ISL_1710849) and B.1.1.317 (S477N, A522S) or PMVL-43 (GISAID EPI_ISL_1710861), B.1.617.2 (T19R G142D E156G F157del R158del L452R T478K D614G P681R D950N) and B.1.617.3 (T19R G142D E156G F157del R158del L452R E484Q D614G P681R D950N) were isolated from a nasopharyngeal swab. 293T/ACE2 or Vero E6 cells were used for isolation and initial passage. SARS-CoV-2 were propagated and titrated on Vero E6 cells. B.1.1.7(hCoV-19/Netherlands/NoordHolland_20432/2020, VOC 202012/01) and B.1.1.28/P.1(hCoV-19/Netherlands/NoordHolland_10915/2021) were obtained from EVAg collection. Viral titers were determined as TCID50 by endpoint dilution assay. Experiments with live SARS-CoV-2 followed the approved standard operating procedures of our biosafety level 3 facility (BSL-3).

### 2.4. Full Genome Viral Sequencing

After two passages on 293T/ACE2, clarified cell supernatants of isolated viruses were used to determine full viral genome sequences. An extraction of total RNA with ExtractRNA Reagent (Eurogen, Moscow, Russia) was performed according to the manufacturer’s instructions. Viral RNA was fragmented and reverse transcribed using random hexamer primers with RevertAid First Strand cDNA Synthesis Kit (ThermoFisher Scientific), followed by dsDNA synthesis using NEBNext Ultra II Non-Directional RNA Second Strand Synthesis Module (NEB). DNA libraries were constructed using NEBNext Fast DNA Library Prep Set for Ion Torrent (NEB) and sequenced using Ion 530 Chip, IonChef instrument and IonTorrent S5XL sequencer (ThermoFisher Scientific, Waltham, MA, USA). Raw reads were quality controlled with vsearch v2.14.2, mapped on the reference Wuhan-Hu-1 sequence (GenBank accession NC_045512.2) with BWA v0.7.17. Consensus sequences were produced using FreeBayes v1.3.2, bcftools v1.9, and bedtools v2.29.2.

### 2.5. RBD Fragment Sequencing

Total RNA was extracted from patient swabs and/or SARS-CoV-2 isolates using the RIBO Prep Kit (FBSI Central Research Institute of Epidemiology of Rospotrebnadzor, Moscow, Russia) according to the manufacturer’s instructions. Amplification was performed using a one-step RT-PCR method based on a reaction mixture containing (for one reaction) 10 pmol of each primer (the forward primer 5′-AACTTTAGAGTCCAACCAACAGAA-3′ and the reverse primer 5′-TGAAGTTGAAATTGACACATTTG-3′), 0.025 mM of each dNTP (Eurogen, Moscow, Russia), and 5 μL of 5X buffer (Eurogen, Moscow, Russia), 200 units of M-MLV reverse transcriptase, 10 units of Taq polymerase, and 10 μL of RNA (appr. 0.5 µg). Oligonucleotides make it possible to obtain an amplicon covering 334- to 529-amino acid positions of the spike-glycoprotein. Amplification was performed on a T100™ Thermal Cycler (Bio-Rad, Hercules, CA, USA). The conditions of the one-step RT-PCR reaction are as follows: 50 °C for 60 min, 95 °C for 5 min, then 35 cycles at 95 °C for 15 s, 55 °C for 10 s and 72 °C for 30 s, then 72 °C for 5 min. The amplification products were purified using ExoSAP-IT™ PCR Product Cleanup Reagent (Thermo Fisher Scientific, Waltham, MA, USA) and the concentration was measured using the Qubit Fluorometer (Thermo Fisher Scientific, Waltham, MA, USA). The obtained fragments were sequenced using the genetic analyzer Applied Biosystems 3500 (Thermo Fisher Scientific, Waltham, MA, USA). The Unipro UGENE v37.0 program was used to analyze the chromatograms of the obtained sequences. Resulting FASTA files were analyzed in Nextclade web-service [17] for the presence of amino acid substitutions, corresponding CSV file was downloaded and analyzed similarly to the GISAID data.

### 2.6. Neutralization Assay with Live SARS-CoV-2

Serum-neutralizing titers of vaccinated against wild-type SARS-CoV-2 variants were analyzed as described earlier [18]. Briefly, serum samples were inactivated by incubation at 56 °C for 30 min. Next, serum was serially diluted in complete Dulbecco’s modified Eagle medium (DMEM) supplemented with 2% fetal bovine serum (FBS) with starting sample dilution at 1:5 with two-fold dilution and mixed with 100 tissue culture infectious dose 50% (TCID50) corresponding SARS-CoV-2 at 1:1 ratio and incubated at 37 °C for 1 h. After that, serum-virus complexes were added to Vero E6 cell monolayer and incubated for 96 h. The cytopathic effect (CPE) of the virus on the cell was assessed visually. Neutralization titer was defined as the highest serum dilution without any CPE in two of three replicable wells.

Determination of NtAb titers was evaluated using the following SARS-CoV-2 variants: B.1.1.1 (PMVL-1, S: D614G; hCoV-19/Russia/Moscow_PMVL-1/2020), B.1.1.7 (hCoV-19/Netherlands/NoordHolland_20432/2020, VOC 202012/01), B.1.351 (hCoV-19/Russia/SPE-RII-27029S/2021), B.1.1.141 (PMVL-31, S: M153T, T385I, D614G; hCoV-19/Russia/MOW-PMVL-31/2020), B.1.1.317 (PMVL-43, S: D138Y, S477N, A522S, D614G, Q675R, A845S; hCoV-19/Russia/MOW-PMVL-43/2021), B.1.1.28/P.1 (hCoV-19/Netherlands/NoordHolland_10915/2021), B.1.617.2 (T19R G142D E156G F157del R158del L452R T478K D614G P681R D950N) and B.1.617.3 (T19R G142D E156G F157del R158del L452R E484Q D614G P681R D950N).

In the study of virus neutralization against wild variants of the SARS-CoV-2 virus, only the serum of vaccinated volunteers without COVID-19 in anamnesis were used. Serum was collected one month after vaccination (after dose 2). All volunteers signed informed consent. Total of 27 sera samples were used for determination of NtAb titer in a comparison study with B.1.1.1, B.1.1.7, B.1.351, B.1.1.141, B.1.1.317 variants and 16 sera samples—in study with B.1.1.1 and B.1.1.28 P.1, B.1.617.2 and B.1.617.3 variants due to the limited volume of 11 sera samples.

Statistical analysis was performed in GraphPad Prism version 9 (GraphPad Software Inc., San Diego, CA, USA). For comparison of paired data, Wilcoxon test was used.

### 2.7. SARS-CoV-2 S Variant Pseudovirus Generation and Neutralization Assay

SARS-CoV-2 pseudotyped particles were produced by transfection of 293T cells using the Transporter 5 transfection reagent as described recently [19]. Briefly, 293T cells were transfected with a mixture 10 μg of pLVPG, 8 μg of pCMV-dR8.2, and 5 µg of pVAX-1-S Wuhan reference strain or pCG1-SARS-2-S bearing the lineages B.1.1.7 and B.1.351 (kindly provided by Dr. Thomas Schultz, Dr. Heino Wiese and Dr. Axel Haverich). Viral supernatants were collected at 72 h after transfection, filtered through a 0.45-μm filter and stored at −80 °C.

Virus neutralization assays were performed on 293T/ACE2 using SARS-CoV-2 spike pseudoviruses that expressed GFP. Pseudotyped virus was incubated with a serial two-fold dilution of human serum samples for 1 h at 37 °C. Then, pseudovirus-serum mixtures were transferred to 293T/ACE2 (1 × 10^4^/well) cells in 96-wells plates. After incubation for 72 h in a 5% CO2 environment at 37 °C, GFP positive cells were counted using a Zeiss Axio Vert.A1 fluorescent microscope (Carl Zeiss AG, Oberkochen, Germany). Neutralization was calculated relative to virus-only controls. The half-maximal neutralization titers (NT50) for serum were determined using nonlinear regression with log (serum dilutions) vs. normalized response (GraphPad Prism).

## 3. Results

Since the beginning of the pandemic, the distribution of genetic lineages in the Russian Federation has notably changed [20,21,22]. In March 2020, the diversity was limited to a few genetic lineages, of which B.1 and B.1.1 were dominant (35.5% and 46.3% respectively). According to the GISAID data, the rate of emergence of new lineages increased during the second wave of pandemic, which began in Russia in September 2020 (Figure 1). Currently the dominant lineages consist of B.1.1, B.1.1.141, B.1.1.336, B.1.1.373 and particularly rising B.1.1.317 and B.1.1.397, which together accounted for 82.4% of the diversity in February 2021.

We then focused on the most common RBD mutations and evaluated their combinations in the main genetic lineages. To rapidly explore the current repertoire of RBD mutations we have developed an in-house Sanger sequencing protocol to detect mutations in the RBD, covering 334–529 amino acid positions of S protein. This protocol was used to sequence 201 samples obtained from the Moscow Infectious Diseases Clinical Hospital patients during the May—June 2020 and November 2020—March 2021 time periods.

The RBD variability data obtained for SARS-CoV-2 variants from Moscow patients (Figure 2, dashed lines) are generally consistent with the variability data for Russian sequences available in GISAID (Figure 2, solid lines and Appendix A) showing increasing prevalence of S477N, A522S, E484K, N501Y, T385I, S494P, N439K, K417N, T487K, N501T, and Y508H mutations. The emergence of mutations in RBD is noted for B.1.1, B.1.1.141, B.1.1.294, B.1.1.317, and B.1.1.397 genetic lineages (Appendix A).

Considering the frequency of individual combinations of RBD mutations we have compiled a list of the most relevant variants (Appendix A). Then, we examined B.1.1.7, B.1.351, and B.1.617, already present in Russia, as well as combinations of variants including B.1.1.317 (S477N, A522S) and B.1.1.141 (T385I) that we were able to isolate using cell culture. All these mutations were previously described in the scientific literature (Appendix A). At the molecular level, the effects of these mutations tend to increase the interaction between the ACE2 receptor and RBD, and/or to reduce the antibodies-neutralizing effect. Although the assessment of VNA using a live virus provides the most reliable data and is the gold standard, for B.1.1.7 and B.1.351 we have further investigated the reduction of serum VNT using live virus and Spike-pseudotyped lentivirus. The live virus neutralization and Spike-pseudotyped lentivirus assays showed concordant results. The study of the neutralizing activity of Sputnik V induced sera against SARS-CoV-2 variants showed no significant differences in the levels of VNT for B.1.1.1, B.1.1.141 (T385I), B.1.1.317 (S477N, A522S), B.1.1.7, and B.1.617.3. About 3.1-, 2.8-, and 2.5-folds decrease in the VNT against B.1.351, P.1, and B.1.617.2 lineages, respectively, was observed (Figure 3).

## 4. Discussion

Epidemiological data are well consistent with the significant role of the ACE2 as an entry receptor. For example, the D614G mutation enhances binding affinity between RBD and ACE2 resulting in increased virus transmissibility [23]. The D614G variants now dominate in most countries worldwide [24]. Fortunately, the increased transmissibility of D614G variants is not associated with increased pathogenicity, nor with antibody-neutralization effects, due to its remoteness from RBD fragment [23]. A slightly more concerning mutation is N439K [25]. Similar to D614G this mutation leads to enhanced binding affinity between RBD and ACE2 but provides resistance to some monoclonal antibodies and eludes some polyclonal responses [25].

Since neither D614G nor N439K has shown the ability to increase the severity of the disease or to escape from neutralizing antibodies, the genetic lineages with these mutations are not classified as VOC. However, the variants of VOC are characterized by RBD mutations, which in turn alter the ACE2 interaction, and diminish the neutralizing antibodies activity.

The most discussed VOC in academic literature and media are Alpha (lineage B.1.1.7), Beta (lineage B.1.351), Gamma (lineage P.1), and Delta (lineage B.1.617.2) and initially found in United Kingdom, South Africa, Brazilia, and India, respectively. The B.1.1.7 lineage was discovered in Kent (UK) in late 2020. There are 23 mutations in the genome B.1.1.7, including a mutation in RBD N501Y, which increases the affinity of binding ACE2 receptor [26]. The spread of the B.1.1.7 lineage was insufficiently constrained by the UK-enforced anti-epidemic measures compared to other genetic variants [27], probably due to a higher basic reproductive number (R0) that increased from 0.4 to 0.7 [28]. There is evidence that B.1.1.7 variants increased lethality [29] and moderate escape from the antibody-neutralizing effects [30,31,32]. This lineage is already dominant in the UK and is widely spread throughout Europe and the United States [28,33,34].

The VNA against B.1.1.7 variant showed 2.1-fold reduction for the AstraZeneca ChAdOx1 nCoV-19 vaccine after 28 days following the second dose compared to SARS-CoV-2 Victoria strain. In a similar study sera VNA levels induced with the Pfizer-BioNTech vaccine BNT162b2 were reduced by 3.3 times [32]. In another study, AstraZeneca vaccine showed nine times VNA reduction in a live virus assay, while the epidemiological decrease of vaccine efficacy was insignificant (not exceeding 11%) [35].

The first case of B.1.1.7 in Russia was officially registered on January 10, 2021, reaching the frequency of 17.4% in March 2021 [36]. In our study, the decrease in VNA efficacy against B.1.1.7 variant for sera of people vaccinated with Sputnik V in both live virus (Figure 3) and Spike-pseudotyped lentiviruses assays (Appendix A) was statistically insignificant. These results demonstrate the high efficacy of the Sputnik V vaccine against the UK variant B.1.1.7, which is actively spreading in Europe, America, and, putatively, Russia.

Recently emerged Delta variant B.1.617 seems to be polymorphic and consists of B.1.617.1, B.1.617.2, and B.1.617.3 [9]. Among these variants B.1.617.2 only has VOC status (VOC-21APR-02). We isolated two variants B.1.617.2 (S: T19R G142D E156G F157del R158del L452R T478K D614G P681R D950N) and B.1.617.3 (S: T19R G142D E156G F157del R158del L452R E484Q D614G P681R D950N). Two mutations of RBD including T478K and E484Q differentiate these variants from each other. Our study reveals that antibody evasion of B.1.617.2 (2.5 VNT decrease) may contribute to the rapid spread of this variant in India and around the world.

The B.1.351 lineage is of much greater concern [37]. This lineage was discovered in South Africa during the first pandemic wave in the heavily affected metropolis (Nelson Mandela Bay) located on the coast of the Eastern Cape province. This genetic lineage spread rapidly and became dominant in the Eastern Cape, Western Cape, and KwaZulu-Natal provinces within several weeks. The available genomic data indicate the rapid spread of this lineage and the displacement of other virus lineages in several regions. This lineage is characterized by eight mutations in S protein, including three substitutions in the RBD (K417N, E484K, and N501Y) resulting in increased transmissibility [37] and immune escape [38]. The separate contribution of these individual mutations was already described [39,40]. Specifically, N501Y is a well-known mutation of the UK strain B.1.1.7 that can increase interaction with ACE2 and contribute to the partial escape from the neutralizing effect of antibodies. For the K417N mutation, a positive effect on the interaction with ACE2 and a weakened interaction with the neutralizing antibody STE90-C11 were shown [40]. The K417N mutation shows a more pronounced effect in combination with N501Y [40] further amplified by the E484K mutation [41,42]. The E484K mutation is currently found in VOC B.1.351, P.1, P.2, and in the newly emerged Alpha strain B.1.1.7 with an additional E484K mutation. A comparative study showed a 20- and 40-fold decrease in viral-neutralizing activity against B.1.351 by sera obtained from the people vaccinated with Moderna/mRNA-1273 and Pfizer/BNT162b2 vaccines, respectively [14]. In the cases of P.1 widely spread in Latin America and Southeast Asia, the reduction in serum neutralization is 4.5 for Moderna/mRNA-1273 and 6.7 for Pfizer/BNT162b2, respectively.

In our study, a three-fold decrease in the viral-neutralizing activity of B.1.351 variant was recorded for sera of the patients vaccinated with Sputnik V for Spike-pseudotyped lentivirus (Appendix A) and a 3.1-fold decrease for a live viral isolate (Figure 3), 2.8-fold decrease was recorded for P.1 variant. The decreased neutralization effect is observed most prominently for weakly reactive serums. Thus, maintaining high titers of antibodies as a result of Sputnik V vaccination or subsequent revaccinations can be one of the solutions to provide high virus-neutralizing activity against VOC B.1.351, P.1, B.1.617.2, and other emerging variants. The results obtained in the current study for Sputnik V compare favorably with the results of other prophylactic vaccines, although direct comparisons are not possible due to the absence of the unified methodology for assessing the virus-neutralizing effect. Single cases of local B.1.351 variant transmissions have been reported in Russia since March 16 [43]. It is noteworthy that, except for B.1.351 has not yet become widespread on other continents, and only imported cases and isolated local transmissions are currently recorded [44]. It is not clear to what extent we can expect this lineage to spread along other continents. The low incidence rate can be attributed both to the insufficient time that has elapsed since the emergence of the B.1.351 lineage, and/or the difficulty of this lineage competing with those already circulating in other continents. It is possible that the competitive advantage can only fully manifest itself after achieving a high percentage of immune people, considering the efficacy at which B.1.351 escapes the neutralizing immune response.

The epidemiologic efficacy of AstraZeneca ChAdOx1 nCoV-19 vaccine against B.1.351 variant was only 10.4% (95% CI, −76.8 to 54.8) [44,45]. It is worth noting that this efficacy is specified for mild to moderate cases of the disease. The protective efficacy of ChAdOx1 nCoV-19 against severe disease is currently unknown. For Novovax recombinant vaccine and Janssen adenovirus vaccine, the epidemiological efficacy in South Africa, where genotype B 1.351 is dominant, was 60% and 57%, respectively [38]. The records for the Janssen vaccine were mostly made for severe disease cases. Probably, most of the vaccines will reduce the preventive effect in the cases of mild disease, though they will retain the preventive effect against cases of severe disease. The problems of recording methodology do not allow for direct comparisons of the epidemiological vaccine performance.

It becomes obvious that under conditions of increasing herd immunity developed as a result of the past disease and vaccination, the number of SARS-CoV-2 variants with mutations in RBD and S protein will rapidly increase. Some of the mutation variants occur independently in different genetic lineages, i.e., N439K [25] or the E484K mutation [9]. In our study, in addition to the emergence of expected VOC in Russia, we studied the diversity of local genetic lineages with mutations in the RBD. A significant percentage of the substitution variants that are becoming common is recorded: S477N + A522S (27.8%), N501Y (4.73%), E484K (3.55%), T385I (2.37%), E484K + S494P (1.77%), N439K (1.18%), Y508H (0.59%), T478K (0.59%), S477N (0.59%), N501T (0.59%). These data are in concordance with previously reported analysis for Russia where S477N, A522S, T385I, and E484K mutation rates were found extensively increasing according to GISAID [10]. All these mutations were already reported as increasing the affinity for ACE2 receptor and/or decreasing the antibody-neutralizing effect (Appendix A). We assessed the virus-neutralizing effect of serum in the patients vaccinated with Sputnik V based on live virus isolates, which showed a non-significant decrease in the antibody-neutralizing effect against B.1.1.397 (T385I) and B.1.1.317 (S477N, A522S).

Virus-neutralizing activity assay protocol used in the current study was previously validated in 1–3 phases of the Sputnik V vaccine clinical trials thus making it possible to directly compare the obtained results with the previous data [1,18].

The virus-neutralizing serum activity is a dynamic parameter. Booster immunization with a second dose is used to increase the number of antibodies and prolong their protective period of action. In case of the Sputnik V, a heterologous prime-booster scheme implementing different vector adenoviruses allows the use of a booster dose as soon as 21 days after the first immunization. In this case, antibodies formed against the adenovirus carrier do not interfere with the boosting effect.

Summarizing the obtained data, we can conclude that for the B.1.1.7, B.1.617.3, and local Moscow variants of genetic lineages B.1.1.141 (T385I) and B.1.1.317 (S477N, A522S) neutralizing properties of Sputnik V induced sera are not changed. For the B.1.351, P.1, and B.1.617.2 VOCs statistically significant 3.1-, 2.8-, and 2.5-fold, respectively, VNA reduction was observed. Notably decrease in VNT to VOC (B.1.351, P.1, B.1.617.2) in Sputnik V vaccinated sera is not as significant as for other vaccines described above. However, in order to make a final conclusion, it is necessary to conduct a direct comparative study. The decreased neutralization effect is of concern and requires further surveillance and epidemiological studies.

## 5. Conclusions

The data obtained indicate no significant differences in VNA against B.1.1.7, B.1.617.3, and local genetic lineages B.1.1.141 (T385I), B.1.1.317 (S477N, A522S) with RBD mutations. For the B.1.351, P.1, and B.1.617.2 statistically significant 3.1-, 2.8-, and 2.5-fold, respectively, VNA reduction was observed. Notably, this decrease is lower than reported in publications for other vaccines. However, a direct comparative study is necessary for a final conclusion.

## Figures and Tables

**Figure 1 vaccines-09-00779-f001:**
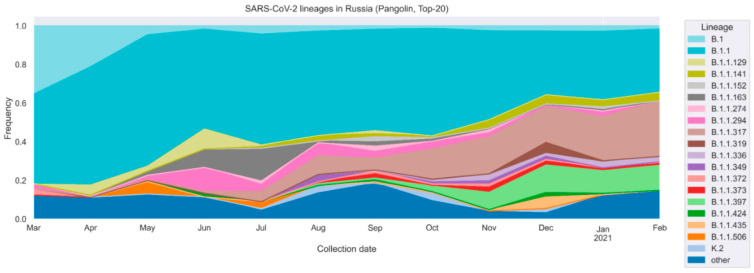
SARS-CoV-2 genetic lineages distribution in Russia. Data were collected from GISAID and visualized to represent 20 most frequent lineages in Russia, PANGOLIN nomenclature.

**Figure 2 vaccines-09-00779-f002:**
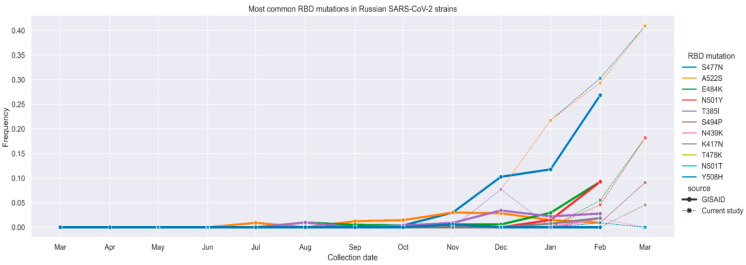
Most common RBD mutations in Russian SARS-CoV-2 variants frequencies of individual mutations according to GISAID data (solid lines) and RBD fragment sequencing data obtained in the current study (dashed lines).

**Figure 3 vaccines-09-00779-f003:**
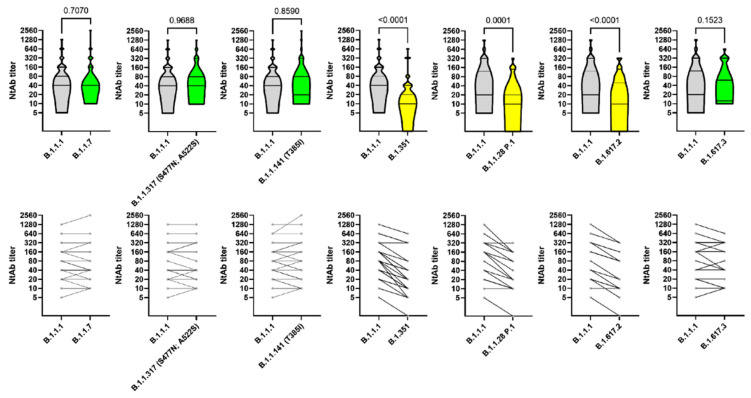
Serum neutralization titers against variants of SARS-CoV-2. The upper graphs show violin plots for serum samples, the lower graphs indicate the individual VNT changes. Geometric mean and 95% CI and p-value (Wilcoxon test) is noted on upper graphs. Number of used serum samples is noted on lower graphs.

## Data Availability

The data presented in this study are available on request from the corresponding author.

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
