# Peer review of "Neutralizing Activity of Sera from Sputnik V-Vaccinated People against Variants of Concern (VOC: B.1.1.7, B.1.351, P.1, B.1.617.2, B.1.617.3) and Moscow Endemic SARS-CoV-2 Variants"

_vaccines, 2021, doi:10.3390/vaccines9070779_

Round 1

Reviewer 1 Report

In the manuscript “Neutralizing activity of sera from Sputnik V vaccinated people against variants of concern (VOC: B.1.1.7, B.1.351, B.1.1.28, B.1.617) and Moscow endemic SARS-CoV-2 variants” the authors state the effects of Sputnik V vaccine  against several SARS-CoV-2 variants.

The subject of this study is interesting and in line with current literature. In general, this study is well conducted, and the paper is very well written with elegant and relevant results in this specific field. 

Author Response

Thank you for your review and appreciation of our work!

Reviewer 2 Report

Gushchin et al describe the study of the neutralization activity of sera from people vaccinated with Sputnik V vaccine against the United Kingdom B.1.1.7, Brazilian 30 B.1.1.28, South African B.1.351, Indian B.1.617 , and Moscow endemic variants B.1.1.141 (T385I) and B.1.1.317 (S477N, A522S). The study shows no significant difference in virus neutralization activity against the UK, Indian (B.1.617.3), and local strains. The virus neutralization activity against the South African variant B.1.351, Brazilian variant B.1.1.28 and Indian B.1.617.2 swere reduced by 3.1- , 2.8- and 2,5- fold, respectively. The data suggest the sera from Sputnik V vaccinated retain neutralizing activity against the UK, Brazilian, South African, Indian, and Moscow local variants. As the mutations on the RBD of these variants increase their transmissibility and escaping from antibody neutralization activity, the study indicates Sputnik V can still protect people from the infection. The manuscript is well written and the results support the conclusion.

Author Response

Thank you for your review and appreciation of our work.

Reviewer 3 Report

The publication “Neutralizing activity of sera from Sputnik V vaccinated people against variants of concern (VOC: B.1.1.7, B.1.351, B.1.1.28, B.1.617) and Moscow endemic SARS-CoV-2 variants” is very timely and topical, it analyses current COVID-19 variants in Moscow (Russia) and also reports on the efficiency of Sputnik V against most infectious variants

 (VOC: B.1.1.7, B.1.351, B.1.1.28, 3 B.1.617) and RGB domain variants detected in Moscow.

The paper is clearly written, very easy to follow, results are supported by sound experiments and conclusions are justified. I would like to suggest the paper to be published after minor revision.

Additional commments

When using PANGOLIN nomenclature please don’t call strains by the names of the countries (For the South 39 African variant B.1.351, Brazilian variant B.1.1.28 and Indian B.1.617.2, say it was discovered first in that country, as you are doing later in the main text.

While including reference in the end of the sentence include dot either before or after the references. Currently this is random.

Abstract:

It will be beneficial if authors do not include abbreviations RGB, VOC ,VNA in the abstract as they are not self-explainable.

Main text

Page 2 line 74  dot is missing after [11],[12].

Page 4 line 168  56°Ð¡ ; extend meaning of “DMEM” and “TCID50”

References

Page 10 lines 452-454 reference 5 is the same as 1 please remove

Page 11 line 508-510 please correct the reference and insert working link.

Supplementary figures

Figure S4, please add the meaning of 001, 002 etc in the figure caption

Author Response

Thank you for your review and appreciation of our work. Your suggestions were taken into account in the text of the paper and SM.

Reviewer 4 Report

This work by Gushchin et al., is a nice piece of work providing added information of the Sputnik V vaccine efficacy for the VOC strains that are emerging as more infectious compared to the initial strains of SARS Cov2. The study is pretty straightforward testing the neutralizing effect of the patient sera vaccinated with Sputnik V on various existing strains.

  1. The work needs some minor experimental proofs to test whether the efficacy of the antibodies increases with certain variations in the neutralizing assays. Varying the time points of neutralizing assay whether that improves the efficacy. Also whether increasing the time difference between dosage improves the efficacy of the vaccine against the newer mutations.
  2. Authors need to profile the antibodies found in the sera of the vaccinated patients and propose what percentage of the antibodies with sequence dominates in them, and vis a vis study the efficacy against the most threatening mutations of those antibodies. This way it would be informative to know the small decrease in VNAs is due to the lacking of which type of antibodies missing. In that way the efficacy of Sputnik V vaccine can be enhanced against these strains.

Minor issues:

1. The authors need to consistent with the use of English format of decimal and European format "." vs ",". This can be improved with a detailed editing of manuscript.

Author Response

Thank you for your review and appreciation of our work. Your suggestions were taken into account in the text of the article.

Regarding your valuable suggestions for further work:

  1. The method for assessing the level of neutralizing antibodies is based on the analysis of inhibition of the cytopathic effect of the SARS-CoV-2 virus on the Vero E6 cell culture. The results are recorded 96 hours after the addition of dilutions of antibodies with the virus to the cells. This time interval allows single infectious particles to induce the development of cytopathic action on cell culture, and we can assess the true level of neutralizing antibodies. Thus, we take the titer of neutralizing antibodies in the final dilution of serum, in which there is a complete suppression of the cytopathic effect of the virus in at least 2 of 3 wells. This technique has been validated and provides consistent results. It has been used in frame in phase 1-3 clinical trials of Sputnik V vaccine. An increase in the incubation time during the analysis can lead to distortion of the test results: obviously higher results of the level of neutralizing antibodies can be obtained. It is important that the same technique is used for all variants of the virus. There is no doubt that it is possible to refine the method for assessing BHA with modification of conditions, some of which can improve the activity of antibodies, but the purpose of this study was to compare the activity of serum against different variants of the virus. That is why it was important for us to use the previously validated methodology that we used in the earlier stages of vaccine development in the 1-3 phase of clinical trials.

    2. Assessment of the profile of antibodies involved in the neutralization of variants of the SARS-CoV-2 virus is an interesting and important fundamental problem. However, our main task was a general assessment of the neutralization of different variants of SARS-CoV-2 in vaccinated volunteers. Currently, the issue of immune protection is extremely acute. Variants of the virus replace each other rapidly and in such a situation it is necessary to understand whether the serum of the vaccinated will be able to neutralize different variants in general, without reference to the antibody profile, in order to promptly inform healthcare and other scientists about the new results. We agree with you that such a study would be very informative and could provide direction for improving vaccines and the basis for further publications.